# A Photoelectrochemical Sensor for the Detection of Hypochlorous Acid with a Phenothiazine-Based Photosensitizer

**DOI:** 10.3390/molecules29030614

**Published:** 2024-01-27

**Authors:** Lijie Luo, Yewen Yang, Shu Chen, Peisheng Zhang, Rongjin Zeng

**Affiliations:** Key Laboratory of Theoretical Organic Chemistry and Functional Molecule of Ministry of Education, School of Chemistry and Chemical Engineering, Hunan University of Science and Technology, Xiangtan 411201, China; 18373877618@163.com (L.L.); 13177365062@163.com (Y.Y.); pshzhang07@gmail.com (P.Z.); zrjxh2@126.com (R.Z.)

**Keywords:** photoelectrochemical sensor, organic photosensitizer, phenothiazine, hypochlorous acid

## Abstract

This paper presents the development of a photoelectrochemical sensor for hypochlorous acid (HOCl) detection, employing a phenothiazine-based organic photosensitizer (Dye-PZ). The designed probe, Dye-PZ, follows a D-π-A structure with phenothiazine as the electron-donating group and a cyano-substituted pyridine unit as the electron-accepting group. A specific reaction of the phenothiazine sulfur atom with HOCl enables selective recognition. The covalent immobilization of Dye-PZ onto a titanium dioxide nanorod-coated fluorine-doped tin oxide electrode (FTO/TiO_2_) using bromo-silane coupling agent (BrPTMS) resulted in the fabrication of the photoanode FTO/TiO_2_/BrPTMS/Dye-PZ. The photoanode exhibited a significant photoresponse under visible-light irradiation, with a subsequent reduction in photocurrent upon reaction with HOCl. The oxidation of the phenothiazine sulfur atom to a sulfoxide diminished the internal charge transfer (ICT) effect. Leveraging this principle, the successful photoelectrochemical sensing of HOCl was achieved. The sensor showed high stability, excellent reproducibility, and selective sensitivity for HOCl detection. Our study provides a novel approach for the development of efficient photoelectrochemical sensors based on organic photosensitizers, with promising applications in water quality monitoring and biosensing.

## 1. Introduction

Hypochlorous acid (HOCl) is a compound renowned for its robust oxidizing capabilities, commonly utilized in household bleach and disinfectants due to its exceptional antimicrobial and stain-removing properties. With oxygen atoms embedded in its molecular structure, HOCl can serve as a potent oxidizing agent and can rapidly react with both organic and inorganic substances. In household cleaning products, HOCl can release reactive oxygen atoms to facilitate the breakdown and removal of colors, making it an indispensable component for domestic and industrial cleaning. Beyond its applications in cleaning, HOCl’s potent disinfecting properties find extensive applications in the medical, food, and water treatment sectors, effectively eliminating various microorganisms to safeguard public health and ensure food safety [1,2,3,4]. At the biological level, HOCl assumes a crucial role, particularly as a representative of reactive oxygen species (ROS). ROS play pivotal roles in physiological and pathological processes, and HOCl, as one of these species, participates in multiple facets of the immune system, contributing significantly to maintaining normal physiological processes [5,6,7,8]. The role of HOCl in immune responses involves pathogen clearance, inflammation promotion, and facilitation of tissue repair, serving as a vital defense mechanism against external threats.

However, excessive exposure to hypochlorous acid (HOCl) can pose severe health risks due to its oxidative and damaging effects on various biological components. The intricate mechanisms underlying HOCl-induced harm involve its reactivity towards crucial cellular structures, leading to widespread physiological disruptions [9,10]. For instance, excess HOCl reacts with cell membrane lipids, causing lipid peroxidation and compromising the integrity of the cell membrane. This may result in increased membrane permeability, disruption of cellular homeostasis, and heightened susceptibility to external threats. Additionally, excess HOCl reacts with thiol-containing amino acids, inducing protein denaturation and functional loss. HOCl also induces DNA strand breaks, base modifications, and cross-linking, impacting nucleic acid integrity and jeopardizing the cell’s ability to maintain genomic stability. Due to these oxidative reactions of HOCl, elevated levels may damage cells and tissues, potentially triggering various diseases such as arthritis, cardiovascular disorders, kidney ailments, neurodegenerative conditions, and cancer [11,12,13,14]. Therefore, detecting the concentration of HOCl is crucial for monitoring its presence in the environment and gaining a deeper understanding of the physiological and pathological functions of HOCl.

Scientists employ a diverse array of methods to monitor and measure HOCl levels in environmental and biological samples, ensuring that it remains within a safe and beneficial range. These techniques include electroanalytical methods [15,16,17], chromatography [18,19], chemiluminescence [20,21,22], colorimetry [23,24,25], and fluorescence [26,27,28,29,30]. Each of these methods is continually evolving for HOCl detection in both scientific research and practical applications. Notably, the focus lies on detection techniques utilizing organic small molecule probes. These molecules, with precise compositions and controllable structures, selectively respond to target analytes through specific functional groups, enhancing selectivity to enable rapid and accurate detection of HOCl in diverse environmental and biological systems. Advancements in detection methods based on the specific induction of oxidation reactions in various structural units by HOCl are ongoing. These structural units include carbon–carbon double bonds, sulfur compounds, aldoxime groups, thioether/Schiff bases, *N*,*N*-dimethylthioamidoformate, phenylboronic acid/borate esters, and others [31,32,33]. Researchers have successfully developed a range of efficient fluorescence probe molecules for HOCl detection by leveraging the specific oxidation reactions in various structural units. Notably, phenothiazine stands out as a crucial structural unit for HOCl recognition. In phenothiazine derivatives, sulfur atoms can be oxidized to sulfoxide by HOCl, achieving a highly specific response to hypochlorous acid [34,35,36,37,38,39,40]. Moreover, phenothiazine compounds exhibit unique biological activities and optoelectronic properties, making them versatile for applications spanning medicine [41,42], photovoltaic cells [43,44,45], fluorescence sensing [46,47,48,49], and electrochemical detection [50,51,52].

In this context, we innovatively employed the phenothiazine structural unit to construct an organic photosensitive small molecule, contributing to the development of a photoelectrochemical (PEC) sensor for detecting HOCl. PEC analysis, utilizing light excitation with electrochemical signals as its output, has gained widespread attention for its improved sensitivity through the utilization of two different signal modes [53,54,55,56,57,58]. Previous research has also indicated that employing organic small-molecule photosensitizers to simultaneously achieve target recognition and photoelectric conversion is an effective strategy for constructing PEC sensors [59,60,61,62,63]. Herein, we firstly synthesized a novel D-π-A structured photoactive molecule (Dye-PZ) by coupling aldehyde-functionalized phenothiazine with cyanopyridine. Subsequently, we covalently immobilized Dye-PZ onto a TiO_2_ nanoarray substrate, obtaining a specific working PEC electrode for detecting HOCl. In Dye-PZ, the phenothiazine moiety serves as the electron-donating group, while the cyanopyridine unit acts as the electron-accepting group. This D-π-A structure exhibits excellent photoelectric conversion efficiency, generating a substantial photocurrent (Figure 1). Upon selective reaction of Dye-PZ with hypochlorous acid, the sulfur in the molecule is oxidized to sulfoxide, disrupting the D-π-A structure and resulting in a decrease in photocurrent response. Consequently, we achieved the highly specific detection of HOCl. Furthermore, this sensor boasts several advantages, such as a fast response, high signal-to-noise ratio, good selectivity, and high sensitivity.

## 2. Results and Discussion

### 2.1. Preparation and Characterization of the FTO/TiO_2_/BrPTMS/Dye-PZ Photoanode

In the design of the photoelectrochemical probe molecule Dye-PZ, the crucial structural unit phenothiazine was incorporated. Phenothiazine serves as the electron-donating moiety, contributing to the construction of a D-π-A structured photoconversion functional molecule. Simultaneously, the sulfur atom in phenothiazine acts as a reactive site for the recognition of the target analyte, hypochlorous acid. The dual functionality of phenothiazine is effectively utilized in the design of Dye-PZ. The synthetic process involves two main steps: the aldehyde functionalization of phenothiazine, followed by condensation with cyano-4-pyridylacetonitrile to yield the target molecule, Dye-PZ. Further details concerning the specific synthetic methodology and characterization are provided in the subsequent experimental section. The results were confirmed through comprehensive characterization using nuclear magnetic resonance (NMR) and mass spectrometry (MS) methods. The photoanode was assembled through two steps (Figure 1). In the first step, the (3-bromopropyl)trimethoxysilane (BrPTMS) was coupled with the FTO/TiO_2_ electrode, resulting in a bromoalkylsilane-modified electrode, FTO/TiO_2_/BrPTMS. In the second step, the alkyl bromide on the modified electrode underwent a covalent reaction with the pyridine nitrogen in Dye-PZ, forming an alkylpyridinium and resulting in photoanode FTO/TiO_2_/BrPTMS/Dye-PZ. To validate the successful assembly of the photoanode, a series of characterization approaches were applied to the electrodes under different modification states.

Firstly, we examined the feasibility of covalent coupling between the bromoalkoxysilane reagent (BrPTMS) and the photoactive molecule (Dye-PZ). As shown in Appendix A, the acetonitrile solution containing only Dye-PZ had an absorption band around 420 nm, while the solution after the reaction with BrPTMS showed absorption around 510 nm. The absorption peak position underwent a redshift, and, simultaneously, the solution color changed from yellow to red. This was attributed to the successful formation of a pyridinium cation through the interaction of Dye-PZ with BrPTMS. The introduction of this pyridinium salt moiety enhances the molecule’s electron-accepting capacity, thereby significantly amplifying the intramolecular photo-induced charge transfer effect (ICT). Consequently, this phenomenon leads to a noticeable redshift in the absorption spectrum. These results indicate the ability of bromoalkoxysilane to form a covalent bond with the photosensitive molecule.

Then, we characterized the morphology of the photoanodes. SEM results revealed a well-ordered TiO_2_ rod array with a radial size of approximately 100 nm grown on the FTO surface (Appendix A). Subsequently, upon modification of the electrode surface with the photoactive molecule Dye-PZ, there was no significant change observed in the surface morphology of the FTO/TiO_2_/BrPTMS/Dye-PZ electrode (Figure 2a). This can be attributed to the relatively smaller size of the organic small molecule, which does not substantially alter the morphology of TiO_2_ upon modification. Next, we characterized the changes in the electrode preparation process. The Fourier-transform infrared spectroscopy (FTIR) outcomes revealed a pronounced absorption band within the 750–650 cm^−1^ range for the TiO_2_ sample, corresponding to the vibration of Ti-O-Ti bonds (Figure 2b, green curve). In the TiO_2_/BrPTMS sample, a subtle absorption band emerged at 1049 cm^−1^, signifying the stretching vibration absorption of the Si-O bond and confirming the interaction between the silane reagent and TiO_2_. In the TiO_2_/BrPTMS/Dye-PZ sample, we observed the aforementioned Ti-O-Ti and Si-O bond absorptions, along with characteristic absorptions of the photoactive molecule (Figure 2b, red curve). The absorption at 1630 cm^−1^ is due to the stretching vibration of the C=N in the pyridinium group formed after the reaction between the bromoalkylsilane reagent and the pyridine moiety of the photoactive molecule. The absorption peak observed at 1560 cm^−1^ corresponds to the stretching vibration of the C=C bond within the pyridine ring. The absorptions at 1460 cm^−1^ and 1340 cm^−1^ correspond to the benzene ring and the C-N stretching vibration in the phenothiazine structure of the photoactive molecule, respectively. These FTIR results indicated that the photoactive molecule Dye-PZ was covalently coupled to the TiO_2_ surface through the silane reagent. Subsequently, the synthesized samples underwent X-ray photoelectron spectroscopy (XPS) elemental analysis. As depicted in Figure 2c, the TiO_2_ sample exhibited peaks at O 1s 529 eV and Ti2p1/2 458 eV. Additionally, due to the organic titanium used in TiO_2_ synthesis, a peak at C 1s 284 eV was observed. In the sample resulting from the reaction of FTO/TiO_2_ with bromosilane, characteristic peaks of Br 3d at 70 eV and Si 2p at 102 eV were evident. Upon covalent coupling with Dye-PZ, new peaks emerged at 163 eV and 399 eV, corresponding to S 2p and N 1s, respectively. The appearance of these peaks signified the successful integration of sulfur and nitrogen components into the sample after coupling with Dye-PZ. These observations further validate the successful synthesis and covalent coupling of the PEC anode FTO/TiO_2_/BrPTMS/Dye-PZ.

Moreover, the electrochemical characterization of the photoanode preparation process was conducted. The electrochemical impedance spectroscopy (EIS) results are depicted in Appendix A. After integrating FTO/TiO_2_ with bromosilane, there was an increase in charge transfer resistance (R_ct_) from 3.1 × 10^3^ Ω to 1.6 × 10^5^ Ω. This increase suggested the effective coverage of the electrode surface by bromosilane, hindering the charge transfer process. Subsequent covalent coupling with Dye-PZ led to a reduction in R_ct_ to 9.1 × 10^4^ Ω. This reduction could be ascribed to the electrostatic attraction between the pyridinium cations and the negatively charged Fe(CN)₆^4−/3−^ redox probes in the electrolyte, leading to a diminished charge transfer resistance. The cyclic voltammetry (CV) results exhibited a similar trend, as shown in Appendix A. These findings provide additional confirmation of the covalent coupling processes, both between FTO/TiO_2_ and BrPTMS and between TiO_2_/BrPTMS and Dye-PZ. This affirmed the successful fabrication of the photoanode.

### 2.2. PEC Response of FTO/TiO_2_/BrPTMS/Dye-PZ to HOCl

To validate the feasibility of the prepared photoanode in responding to HOCl, we initially tested the spectral response of the synthesized photoactive probe Dye-PZ to HOCl in solution. HPLC-MS experiments were conducted using different solution systems to verify the occurrence of the reaction. Subsequently, we employed a three-electrode system to assess the PEC response of the photoanode to HOCl. In a mixture of DMSO and phosphate buffer (*v*/*v*, 5/5, 5 mM, pH 7.4), the spectral response of Dye-PZ to HOCl was investigated. Dye-PZ (10 μM) exhibited a blue shift in the UV absorption spectrum from 422 nm to 400 nm upon reacting with HOCl. Simultaneously, a significant enhancement in fluorescence emission at 470 nm was observed, and, under UV light, the solution’s fluorescence changed from red to blue (Figure 3). The spectral response of Dye-PZ to HOCl can likely be attributed to the oxidation of the sulfur in the phenothiazine moiety induced by HOCl, leading to the attenuation of the intra-molecular charge transfer (ICT) effect, resulting in spectral shifts. Subsequently, the reaction process was monitored using HPLC-MS (Appendix A). A chromatographic peak appeared at 5.14 min for Dye-PZ alone (Appendix A), while after reacting with HOCl, two peaks emerged at 5.14 min and 5.83 min (Appendix A), corresponding to Dye-PZ (*m*/*z* [M + H^+^]^+^: theoretical value 342.1, measured value *m*/*z*: 342.1) and the product (*m*/*z* [M + H^+^]^+^: theoretical value 358.1, measured value *m*/*z*: 358.1), respectively. These results indicated the oxidation of sulfur in Dye-PZ to sulfoxide after reacting with HOCl.

Following that, the PEC response of the photoanode to HOCl was tested. As depicted in Figure 1, the phenothiazine moiety within the molecule functioned as the electron-donating group, while the pyridine unit served as the electron-accepting group, inducing the generation of the ICT effect and absorption in the visible region. Upon reacting with HOCl, the sulfur in the molecule underwent oxidation to a sulfoxide, leading to the elimination of the ICT effect. The color of the photoanode FTO/TiO_2_/BrPTMS/Dye-PZ also changed from red to colorless (Figure 4 inset). Photocurrent tests demonstrated stable photocurrent responses under 500 nm excitation light. The process of photocurrent generation begins with the excitation of Dye-PZ under illumination, where electrons transition from the HOMO state to the LUMO excited state. Subsequently, the injected electrons from the LUMO into the covalently coupled TiO_2_ conduction band are collected by the electrode, generating a photocurrent signal. The oxidized state of Dye-PZ, after losing electrons, is then reduced to its initial state by an electron-donating species in the solution, such as ascorbic acid, thereby continuing the photocurrent generation. However, upon reacting with HOCl, the molecular structure of the D-π-A system is disrupted, preventing the effective generation of photocurrent. This phenomenon can be observed as a significant decrease in photocurrent in Figure 4. The results confirm the effectiveness of the photoelectrode as a PEC sensing platform for HOCl.

### 2.3. Analytical Performance

During PEC testing, factors such as bios voltage, the excitation light, and the electron donor can influence the photocurrent response. Therefore, we explored and optimized the conditions of this method to obtain the best experimental results. Firstly, we investigated the impact of bias voltage (−0.6 V to −0.2 V) and excitation wavelength (440–600 nm) on the PEC response. As shown in Appendix A, the photoanode exhibited a stable response and significant current changes at −0.4 V (vs. Hg/Hg_2_SO_4_) and 500 nm excitation wavelength. Additionally, three electrolytes containing ascorbic acid (AA), sodium oxalate, and triethanolamine (TEOA) as electron donors were chosen for comparative testing to explore the influence of different electron donors on PEC testing. The results in Appendix A indicate that the photocurrent response showed the largest relative reduction when ascorbic acid was used as the electron donor. Therefore, we selected a −0.4 V bias voltage, a 500 nm excitation light, and an electrolyte-containing ascorbic acid for subsequent testing. Under the optimal conditions, the photoanode was tested for its photocurrent response to different concentrations of HOCl (Figure 5a). We observed that within the concentration range of 5.0–40.0 μM HOCl, the photocurrent exhibited a decrease as the concentration increased, showing a robust linear correlation with the logarithm of HOCl concentration, as illustrated in Figure 5b, represented by the fitting equation I = −0.348 × lg[HOCl] + 0.861 (R^2^ = 0.981). The detection limit was established as 2.58 μM (3σ/k).

Next, the selectivity and anti-interference performance of the PEC sensor were investigated. Interfering ions included common ions such as Al^3+^, Fe^2+^, Cu^2+^, Zn^2+^, Mg^2+^, Mn^2+^, NO_3_^−^, Cl^−^, BrO_3_^−^, SO_4_^2−^, H_2_O_2_, C_2_H_5_OH, and alanine (Ala). None of the mentioned interfering substances led to a substantial reduction in the PEC sensor’s photocurrent. Although there are studies reporting that thiol-containing compounds (such as dimethyl sulfide, DMS) can be oxidized to sulfoxides by hydrogen peroxide [64,65], in the case of this Dye-PZ probe, the sulfur (S) in phenothiazine is linked to two phenyl rings, with its electron cloud distributed in a larger conjugated system. Therefore, the reactivity of sulfur in phenothiazine is expected to be lower than that of alkyl thioethers, making it resistant to oxidation by hydrogen peroxide and selectively oxidizable by the stronger oxidizing agent, hypochlorous acid, thus achieving the interference-free detection of hydrogen peroxide. Phenothiazine is also widely utilized as a high-specificity recognition moiety in constructing probes for hypochlorous acid, and these probe molecules have also demonstrated interference-free characteristics against hydrogen peroxide [66,67,68,69,70]. Furthermore, competitive experiments revealed that the presence of other interferences did not impact the detection of HOCl (Figure 6a). The selected competitive results indicated that the PEC sensor exhibited outstanding anti-interference capability. Finally, the stability of the sensor was examined by testing its continuous PEC pulse response. After eight cycles of testing, the photocurrent remained highly stable, with a relative standard deviation (RSD) of less than 1.1% (Figure 6b), demonstrating the sensor’s high photostability.

## 3. Materials and Methods

### 3.1. Materials and Apparatus

The materials for synthesizing the photoactive molecule Dye-PZ, including 10-methyl-10H-phenothiazine, phosphorus oxychloride, cyanopyridine, and piperidine, were purchased from Shanghai Titan Technology Co., Ltd. (Shanghai, China). Titanium butoxide and acetonitrile were procured from Tianjin Chemreagent Company (Tianjin, China). Solvents used in synthesis, such as dichloromethane and ethyl acetate, were obtained from Tianjin Damao Chemical Reagent Factory. Analytical-grade small molecules and salts were sourced from China National Pharmaceutical Group Corporation. All reagents were of analytical purity. Deionized water used for preparing testing solutions was purified through a secondary deionization process.

UV–visible spectrophotometry signals were recorded using a Lambda-850 instrument from PerkinElmer Inc. (Waltham, MA, USA). Fluorescence measurements were conducted with an RF-5301PC fluorometer from Shimadzu Corporation (Kyoto, Japan). Mass spectrometry analysis employed a Q-Tof Premier mass spectrometer from Waters Corporation (Milford, MA, USA). Nuclear magnetic resonance (NMR) spectra were acquired using a Bruker ARX-400 spectrometer (Bruker Limited, Fällanden, Switzerland). Infrared spectra were recorded with the Bruker Vector 22 Fourier-transform infrared spectrometer (Bruker Limited, Switzerland). X-ray photoelectron spectroscopy (XPS) measurements were conducted using a Thermo Scientific K-Alpha instrument (Thermo Fisher Scientific Inc., Waltham, MA, USA). Photoelectrochemical tests were carried out using a DY200B electrochemical workstation from Digi-Ivy, Inc. (Austin, TX, USA).

### 3.2. Synthesis of Dye-PZ

The synthetic route for Dye-PZ is depicted in Appendix A. Under an ice bath (0 °C) and a nitrogen atmosphere, dimethylformamide (0.17 mL, 2.4 mmol) was combined with phosphorus oxychloride (0.2 mL, 2.4 mmol), followed by stirring for 15 min. Subsequently, compound **1** (426 mg, 2 mmol in 2 mL anhydrous DMF) was introduced into the mixture. The reaction mixture was heated to 60 °C and stirred for 4 h. After completing the reaction, the mixture was transferred into 100 mL of ice water and neutralized with a 10% NaHCO_3_ solution. The resulting viscous material was subjected to three extractions with 100 mL of dichloromethane. The organic layer was separated, washed sequentially with saturated sodium chloride solution and water, dried over anhydrous Na_2_SO_4_, and the solvent was evaporated under reduced pressure. The residue was further purified through silica gel column chromatography, using dichloromethane as the eluent. This process resulted in the isolation of purified compound **2** (0.322 g, 67% yield). Compound **2** (241 mg, 1.0 mmol), cyanopyridine (118 mg, 1.0 mmol), and 4 mL anhydrous ethanol were combined and refluxed for 5 h. After the completion of the reaction, the solvent was removed under reduced pressure, and further purification was carried out using silica gel column chromatography. A mixture of dichloromethane and ethyl acetate (1:1, *v*/*v*) was used as the eluent for purification. The resulting product, Dye-PZ, was obtained as a deep red solid (0.143 g, 43% yield). Characterization of the synthesized photosensitizer Dye-PZ was conducted using several techniques, including nuclear magnetic resonance (NMR) spectroscopy and mass spectrometry. ^1^HNMR (400 MHz, DMSO) δ 8.67 (d, J = 6.1 Hz, 2H), 8.19 (s, 1H), 7.91 (dd, J = 8.6, 1.9 Hz, 1H), 7.81 (d, J = 1.9 Hz, 1H), 7.69 (d, J = 6.1 Hz, 2H), 7.26 (t, J = 7.7 Hz, 1H), 7.20 (d, J = 7.4 Hz, 1H), 7.13 (d, J = 8.7 Hz, 1H), 7.03 (t, J = 7.3 Hz, 2H), 3.39 (s, 3H) (Appendix A). ^13^CNMR (101 MHz, DMSO) δ 150.85, 148.41, 144.87, 144.29, 141.84, 130.84, 128.55, 128.09, 127.66, 127.40, 123.85, 122.68, 121.48, 119.95, 117.96, 115.71, 115.18, 104.96, 35.91 (Appendix A). MS [ESI]: *m*/*z*, theoretical molecular weight [M^+^]^+^ 341.0981; mass spectrum data 341.0983 (Appendix A).

### 3.3. Preparation of FTO/TiO_2_/BrPTMS/Dye-PZ

The FTO/TiO_2_ nanorod-based electrode was prepared according to a reported method [60]. The assembly of the photoanode is illustrated in Figure 1, and the specific assembly steps were as follows: Firstly, 3-(bromopropyl)trimethoxysilane (150 μL) and isopropanol (10 mL) were added to a beaker. After thorough mixing, the prepared FTO/TiO_2_ was placed into the beaker and the reaction mixture was heated at 70 °C for 30 min. Subsequently, the electrode was removed and washed with isopropanol, and dried with nitrogen gas. Then, the electrode was placed in a vacuum drying oven at 60 °C to obtain the bromosilane-modified electrode FTO/TiO_2_/BrPTMS. Next, FTO/TiO_2_/BrPTMS was immersed in a solution containing Dye-PZ (1 mM) in acetonitrile (20 mL) at 80 °C for 5 h oil bath reflux. Finally, the electrode was thoroughly rinsed with dimethyl sulfoxide to remove non-covalently bound dye, resulting in the desired working electrode FTO/TiO_2_/BrPTMS/Dye-PZ.

### 3.4. Characterization and Photoelectrochemical Testing of FTO/TiO_2_/BrPTMS/Dye-PZ

The photoanode and its preparation process were characterized using a three-electrode system with K_3_Fe(CN)₆ (10 mM) and KCl (0.1 M) solution as the electrolyte. The three electrodes consisted of differently modified FTO as the working electrode (WE), Ag/AgCl (saturated KCl at 25 °C) as the reference electrode (RE), and platinum wire as the counter electrode (CE). The photoanode was subjected to electrochemical impedance spectroscopy (EIS) and cyclic voltammetry (CV) using a CHI 660E electrochemical analyzer. Infrared spectroscopy (IR) was employed for functional group analysis, and X-ray photoelectron spectroscopy (XPS) was used for elemental analysis of the photoanode surface.

The photoanode functioned as the working electrode, accompanied by a platinum wire as the counter electrode and Hg/Hg_2_SO_4_ (saturated K_2_SO_4_ at 25 °C) as the reference electrode. Incubation took place in a phosphate buffer solution (5 mM, pH 7.4) containing a specific concentration of HOCl. The electrolyte utilized was a phosphate buffer solution (0.1 M, pH 7.4) comprising 0.1 M sodium sulfate and 0.1 M ascorbic acid (AA). The photoelectrochemical (PEC) characteristics were monitored using an electrochemical workstation and a fluorescence spectrophotometer (model: RF-5301PC, Shimadzu, Japan) equipped with a xenon lamp for excitation.

## 4. Conclusions

A PEC sensor for HOCl detection was successfully developed using a phenothiazine-based photoactive probe. The designed probe, Dye-PZ, featuring a D-π-A structure with phenothiazine as the electron-donating group and a cyano-pyridine unit as the electron-withdrawing group, demonstrated the specific reactivity with HOCl. Covalent attachment of Dye-PZ onto a titanium dioxide nanorod substrate using BrPTMS yielded the photoanode FTO/TiO_2_/BrPTMS/Dye-PZ. The photoanode exhibited a pronounced PEC response under visible light, and, upon reaction with HOCl, the oxidation of the sulfur atom in the phenothiazine resulted in a weakened internal charge transfer effect and a subsequent decrease in photocurrent. Exploiting this mechanism, the PEC sensor demonstrated effective detection of HOCl. The sensor displayed high stability and excellent reproducibility, allowing for the selective and sensitive detection of HOCl.

## Data Availability

Data are contained within the article and Appendix A.

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
