# Peer review of "A Photoelectrochemical Sensor for the Detection of Hypochlorous Acid with a Phenothiazine-Based Photosensitizer"

_molecules, 2024, doi:10.3390/molecules29030614_

Round 1

Reviewer 1 Report

Comments and Suggestions for Authors

It's good work that may interest the scientists working in the field.

Some minor corrections are needed.

There is no mention of Fig. 2 in the text, and its caption is general. Please indicate what shows 2a, 2b, and 2c.

In the text, there is “Subsequently, the reaction process was 184 monitored using HPLC-MS (Figure S3),” but Fig. S3 shows cyclic voltammetry.

Please give the yield of compound 2 and the eluent of the silica gel chromatography.

Please give the eluent of the silica gel chromatography for the Dye-HOCl.

I propose to accept it for publication with minor changes

Comments on the Quality of English Language

No comments

Author Response

Reviewers’ comments

Reviewer 1

It's good work that may interest the scientists working in the field.

Some minor corrections are needed.

Comment 1: There is no mention of Fig. 2 in the text, and its caption is general. Please indicate what shows 2a, 2b, and 2c.

Response: We sincerely appreciate the valuable feedback from the reviewer. We have now appropriately referenced Figure 2 in the manuscript, including specific mentions of 2a, 2b, and 2c.

Comment 2: In the text, there is “Subsequently, the reaction process was 184 monitored using HPLC-MS (Figure S3),” but Fig. S3 shows cyclic voltammetry.

Response: Thank you for pointing out the mislabeling. It should indeed refer to Figure S4, and we have made the necessary correction.

Comment 3: Please give the yield of compound 2 and the eluent of the silica gel chromatography.

Response: We have supplemented the experimental section with the yield of compound 2 and the eluent used for silica gel chromatography.

Comment 4: Please give the eluent of the silica gel chromatography for the Dye-HOCl.

Response: We have supplemented the experimental section with the eluent used for the silica gel chromatography of the separated probe molecule.

Reviewer 2 Report

Comments and Suggestions for Authors

In the manuscript titled “A Photoelectrochemical Sensor for the Detection of Hypochlorous Acid with a Phenothiazine-based Photosensitizer”, Lijie Luo and coworkers evaluated a photoelectrochemical sensor for HOCl detection. A photoanode sensor composed of a donor-π-acceptor molecule comprised by a phenothiazine donor and a cyano-substituted pyridinium acceptor was immobilized onto a bromoalkylsilane-modified FTO/TiO2 electrode. In DMSO/phosphate buffer 5/5 mixture solution, the donor-π-acceptor chemosensor displayed hypsochromic shift of UV-vis bands and prominent fluorescence enhancement accompanied by a change in fluorescence color from red to blue in the presence of HOCl.

The authors explained this observation in terms of switching off the internal charge transfer (ICT) effect as plausible mechanism, due to oxidation reaction of sulfide to sulfoxide in phenoxythiazine group by HOCl. The authors demonstrated the selectivity of the PEC sensor from other different ions, as well as its highly stability and robustness.

The work presented in the manuscript is of high interest to the broad and interdisciplinary readership of Molecules journal. A similar work has been recently published by the authors (see Cheng, J.; Luo, Y.; Hao, Y.; Han, H.; Hu, X.; Yang, Y.; Long, X.; He, J.; Zhang, P.; Zeng, R.; Xu, M.; Chen, S. A Responsive Organic Probe Based Photoelectrochemical Sensor for Hydrazine Detection. Spectrochim. Acta Part A: Mol. Biomol. Spectrosc. 2024, 305, 123463. https://doi.org/10.1016/j.saa.2023.123463). The manuscript is well written, and presentation meets high quality standards of the journal.

The following minor revisions must be addressed before the manuscript can be accepted.

1. It is shown that addition of HOCl to PEC sensor diminishes the photocurrent (Figure 5a); where the photocurrent decreases with as little as 5.0 μM HOCl. This is not the case in Figure 6, where an increase in photocurrent of blank or interfering ion solution (red column) upon addition of 60 μM of HOCl (black column), from ca. 0.2 μA up to ca. 0.6 μA was observed. The authors should give an explanation to that observation.

2. Can the authors comment on the selectivity or anti-interference mechanism when a strong oxidant such as H2O2 was evaluated, since it would oxidize sulfide into sulfoxide as in the case of HOCl (see Jhih Wei Chu and Bernhardt L. Trout On the Mechanisms of Oxidation of Organic Sulfides by H2O2 in Aqueous Solutions J. Am. Chem. Soc., 2004, 126(3), 900‒908. https://doi.org/10.1021/ja036762m).

3. Compound characterization and identification must comply with author guidelines, therefore High Resolution Mass Spectrometry (HRMS) or elemental analysis of the final compound should be included in the supporting information.

4. A general paragraph that describes the acquisition of synthesis materials and general analytical approaches employed in the study in section “3. Materials and Methods” should be included.

Minor corrections

The abbreviated name given to the final compound, namely Dye-HOCl is confusing since there is no HOCl molecule as part of the chemical structure. Therefore, it is recommended to name it in a different way, removing the HOCl part.

Section 2 should be named “2. Results and discussion”

Line 106, page 3 – should be rewritten as “forming an alkylpyridinium”.

Line 185, page 5 – should be Figure S4 instead of Figure S3.

The equation in line 227, page 6 should be corrected as I = −0.348 × log[HOCl] + 0.861. A similar correction should be done in Figure 5b (page 6), x axis label should be written as log([HOCl]) without units.

In line 276, page 8, “reportedmethod” must be corrected.

Correct the sentence “for 5 hunder oil bath” in line 284, page 8.

Define compound “AA” in line 301, page 8.

Correct “substrateusinig” in line 310, page 8.

If possible, references should include DOI.

Comments on the Quality of English Language

Minor English revision must be performed by the authors. Next are some issues found regarding quality of English language:

Sentences in line 196, page 5 and line 233, page 6 are not clear and should be rewritten.

Past tense must be used for the methodology description in section 3.2 Preparation of FTO/TiO2/BrPTMS/Dye-HOCl, lines 275 to 286, pages 7 and 8.

Author Response

Reviewer 2

In the manuscript titled “A Photoelectrochemical Sensor for the Detection of Hypochlorous Acid with a Phenothiazine-based Photosensitizer”, Lijie Luo and coworkers evaluated a photoelectrochemical sensor for HOCl detection. A photoanode sensor composed of a donor-π-acceptor molecule comprised by a phenothiazine donor and a cyano-substituted pyridinium acceptor was immobilized onto a bromoalkylsilane-modified FTO/TiO2 electrode. In DMSO/phosphate buffer 5/5 mixture solution, the donor-π-acceptor chemosensor displayed hypsochromic shift of UV-vis bands and prominent fluorescence enhancement accompanied by a change in fluorescence color from red to blue in the presence of HOCl.

The authors explained this observation in terms of switching off the internal charge transfer (ICT) effect as plausible mechanism, due to oxidation reaction of sulfide to sulfoxide in phenoxythiazine group by HOCl. The authors demonstrated the selectivity of the PEC sensor from other different ions, as well as its highly stability and robustness.

The work presented in the manuscript is of high interest to the broad and interdisciplinary readership of Molecules journal. A similar work has been recently published by the authors (see Cheng, J.; Luo, Y.; Hao, Y.; Han, H.; Hu, X.; Yang, Y.; Long, X.; He, J.; Zhang, P.; Zeng, R.; Xu, M.; Chen, S. A Responsive Organic Probe Based Photoelectrochemical Sensor for Hydrazine Detection. Spectrochim. Acta Part A: Mol. Biomol. Spectrosc. 2024, 305, 123463. https://doi.org/10.1016/j.saa.2023.123463). The manuscript is well written, and presentation meets high quality standards of the journal.

The following minor revisions must be addressed before the manuscript can be accepted.

Comment 1: It is shown that addition of HOCl to PEC sensor diminishes the photocurrent (Figure 5a); where the photocurrent decreases with as little as 5.0 μM HOCl. This is not the case in Figure 6, where an increase in photocurrent of blank or interfering ion solution (red column) upon addition of 60 μM of HOCl (black column), from ca. 0.2 μA up to ca. 0.6 μA was observed. The authors should give an explanation to that observation.

Response: We sincerely appreciate the positive evaluation and valuable feedback from the reviewer. Thank you for the meticulous review and pointing out the error. There was a labeling issue in the initial version of Figure 6; the black column represents the photocurrent of the blank electrode and the electrode reacting with the interfering substance, while the red column represents the current response after adding hypochlorous acid in the presence of the interfering substance. We have corrected this error in the revised manuscript.

Comment 2: Can the authors comment on the selectivity or anti-interference mechanism when a strong oxidant such as H2O2 was evaluated, since it would oxidize sulfide into sulfoxide as in the case of HOCl (see Jhih Wei Chu and Bernhardt L. Trout On the Mechanisms of Oxidation of Organic Sulfides by H2O2 in Aqueous Solutions J. Am. Chem. Soc., 2004, 126(3), 900‒908. https://doi.org/10.1021/ja036762m).

Response: Thank you to the reviewer for providing relevant reference. The mentioned paper utilizes theoretical calculations and experimental methods to explore the process of hydrogen peroxide (H2O2) oxidizing dimethyl sulfide (DMS) into dimethyl sulfoxide. The results indicate that under moderate pH conditions, between 2 and 12, H2O2 undergoes O-O bond dissociation and forms an S-O bond, leading to the oxidation of dimethyl sulfide. In our study, we use the phenothiazine structural unit as a recognition group for the response to hypochlorous acid. Although phenothiazine contains a sulfur atom, and the sulfur atom is the reactive site for hypochlorous acid, the S atom in this structure is connected to two benzene rings, and its electron cloud distribution is within a larger conjugated system. Therefore, the reactivity of sulfur (S) in phenothiazine is expected to be lower than the dimethyl sulfide reactivity mentioned in the literature (J. Am. Chem. Soc., 2004, 126, 900). As a result, it can resist oxidation by hydrogen peroxide and be selectively oxidized by the stronger oxidizing agent, hypochlorous acid, enabling interference-free detection of hydrogen peroxide. This achieves selective detection without interference from hydrogen peroxide. Phenothiazine has been widely used as a highly specific recognition group to construct hypochlorous acid probes (Chem. Commun., 2015, 51, 1442; Adv. Funct. Mater., 2017, 27, 1700493; Chem.-Eur. J. 2018, 24, 8157; Anal. Chem. 2022, 94, 2, 811). The probes in these reports also demonstrates resistance to interference from hydrogen peroxide. Based on the reviewer's suggestion, we have included relevant discussions in the selectivity results section.

Comment 3: Compound characterization and identification must comply with author guidelines, therefore High Resolution Mass Spectrometry (HRMS) or elemental analysis of the final compound should be included in the supporting information.

Response: We have supplemented the High-Resolution Mass Spectrometry results for the final product probe.

Comment 4: A general paragraph that describes the acquisition of synthesis materials and general analytical approaches employed in the study in section “3. Materials and Methods” should be included.

Response: In the experimental section, we have added a paragraph describing the acquisition of synthesis materials and the general analytical approaches employed in the study.

Comment 5: Minor corrections

The abbreviated name given to the final compound, namely Dye-HOCl is confusing since there is no HOCl molecule as part of the chemical structure. Therefore, it is recommended to name it in a different way, removing the HOCl part.

Section 2 should be named “2. Results and discussion”

Line 106, page 3 – should be rewritten as “forming an alkylpyridinium”.

Line 185, page 5 – should be Figure S4 instead of Figure S3.

The equation in line 227, page 6 should be corrected as I = −0.348 × log[HOCl] + 0.861. A similar correction should be done in Figure 5b (page 6), x axis label should be written as log([HOCl]) without units.

In line 276, page 8, “reportedmethod” must be corrected.

Correct the sentence “for 5 hunder oil bath” in line 284, page 8.

Define compound “AA” in line 301, page 8.

Correct “substrateusinig” in line 310, page 8.

If possible, references should include DOI.

Sentences in line 196, page 5 and line 233, page 6 are not clear and should be rewritten.

Past tense must be used for the methodology description in section 3.2 Preparation of FTO/TiO2/BrPTMS/Dye-HOCl, lines 275 to 286, pages 7 and 8.

Response: We appreciate the detailed and important feedback from the reviewer. We have made revisions based on the reviewer's suggestions, and the modified content is highlighted in yellow in the manuscript.

Reviewer 3 Report

Comments and Suggestions for Authors

The manuscript "A Photoelectrochemical Sensor for the Detection of Hypochlorous Acid with a Phenothiazine-based Photosensitizer" describes the fabrication of the new PES sensitive to hypochlorite ions, which functions because of the ClO--induced oxidation of phenothiazine-based molecule linked to the surface of TiO2 through 3-bromopropyl)trimethoxysilane. Each step of the paper is carefully justified by the Authors: they consequently give the evidence for the organic probe attached to the surfase of TiO2; study the mechanism of ClO- action on the organic molecule demostrating that the oxidation occurs, which leads to the naked-eye visible changes in UV-Vis spectra and photoluminescence of the probe; finally, they thoroughly study the electric current response of the modified TiO2 plate upon action of ClO- ions deducing the linear dependence between loss in the current and hypochlorite concentration. The sensor developed is selective and stable as was also shown. It can be easily used in daily practice to control the presence of hazardous hypochlorite ions in water.

Limitations of the work: I believe that other reducing agents can mask ClO- anions in the solutions if their formal electrode potential exceeds that of the phenothiazine derivative used. In addition, such anions as Br-, I- will also interfere hypochlorite determination because of the chemical reactions between halides and hypochlorite. Moreover, it would be interesting to study the interfering action of such mixures as Fe2+ + H2O2 or Cu2+ + H2O2 capable of ROS generating due to the Fenton reaction.

The above-mentioned limitations, however, do not dminish the advantages proposed by Authors in their paper.

Author Response

Reviewer 3

The manuscript "A Photoelectrochemical Sensor for the Detection of Hypochlorous Acid with a Phenothiazine-based Photosensitizer" describes the fabrication of the new PES sensitive to hypochlorite ions, which functions because of the ClO--induced oxidation of phenothiazine-based molecule linked to the surface of TiO2 through 3-bromopropyl)trimethoxysilane. Each step of the paper is carefully justified by the Authors: they consequently give the evidence for the organic probe attached to the surfase of TiO2; study the mechanism of ClO- action on the organic molecule demostrating that the oxidation occurs, which leads to the naked-eye visible changes in UV-Vis spectra and photoluminescence of the probe; finally, they thoroughly study the electric current response of the modified TiO2 plate upon action of ClO- ions deducing the linear dependence between loss in the current and hypochlorite concentration. The sensor developed is selective and stable as was also shown. It can be easily used in daily practice to control the presence of hazardous hypochlorite ions in water.

Limitations of the work: I believe that other reducing agents can mask ClO- anions in the solutions if their formal electrode potential exceeds that of the phenothiazine derivative used. In addition, such anions as Br-, I- will also interfere hypochlorite determination because of the chemical reactions between halides and hypochlorite. Moreover, it would be interesting to study the interfering action of such mixures as Fe2+ + H2O2 or Cu2+ + H2O2 capable of ROS generating due to the Fenton reaction.

The above-mentioned limitations, however, do not dminish the advantages proposed by Authors in their paper.

Response: We greatly appreciate the reviewer's positive assessment of our work. Phenothiazine is a crucial recognition unit for designing probes for hypochlorite detection. In previous studies involving fluorescence probe molecules, this recognition group has exhibited outstanding selectivity. In our work, we employ phenothiazine both as a recognition unit for the photoelectrochemical probe molecule and as a component for constructing photoresponsive molecules (phenothiazine is also utilized as an electron donor to construct photoelectric conversion molecules). It has demonstrated good selectivity, showing resistance to the influence of oxidizing substances such as bromide ions and not responding to hydrogen peroxide (H2O2). The reviewer raises an excellent question regarding the stability and selectivity of the probe molecule in complex redox systems. Unfortunately, due to time constraints, we were unable to conduct a systematic investigation. In our future, more in-depth research, we will address this issue and explore the stability and selectivity of the probe in complex redox environments.